# Cellular variability of nonsense-mediated mRNA decay

Hanae Sato[1] & Robert H. Singer ⬤ [1,2,3✉]

Nonsense-mediated mRNA decay (NMD) is an mRNA degradation pathway that eliminates transcripts containing premature termination codons (PTCs). Half-lives of the mRNAs containing PTCs demonstrate that a small percent escape surveillance and do not degrade. It is not known whether this escape represents variable mRNA degradation within cells or, alternatively cells within the population are resistant. Here we demonstrate a single-cell approach with a bi-directional reporter, which expresses two β-globin genes with or without a PTC in the same cell, to characterize the efficiency of NMD in individual cells. We found a broad range of NMD efficiency in the population; some cells degraded essentially all of the mRNAs, while others escaped NMD almost completely. Characterization of NMD efficiency together with NMD regulators in single cells showed cell-to-cell variability of NMD reflects the differential level of surveillance factors, SMG1 and phosphorylated UPF1. A single-cell fluorescent reporter system that enabled detection of NMD using flow cytometry revealed that this escape occurred either by translational readthrough at the PTC or by a failure of mRNA degradation after successful translation termination at the PTC.

[1] Department of Anatomy and Structural Biology, Albert Einstein College of Medicine, 1300 Morris Park Ave, Bronx, NY 10461, USA. [2] Department of Cell Biology, Albert Einstein College of Medicine, 1300 Morris Park Ave, Bronx, NY 10461, USA. [3] Gruss Lipper Biophotonics Center, Albert Einstein College of Medicine, 1300 Morris Park Ave, Bronx, NY 10461, USA. ✉email: robert.singer@einsteinmed.edu

Nonsense-mediated mRNA decay (NMD) is a highly conserved mRNA degradation pathway in eukaryotes. NMD eliminates aberrant mRNAs containing premature termination codons (PTC) that potentially produce truncated dominant-negative or gain-of-function proteins[1] as well as normal transcripts containing cis-elements such as introns in their 3′ untranslated region (UTR) or longer 3′UTRs[2–4]. A major research focus has been to understand how NMD recognizes its target transcripts. In mammalian cells, two models have been suggested to explain selective mRNA degradation in NMD: the "EJC-model", whereby the existence of splicing-generated exon-junction complexes (EJCs) more than 50–55 nucleotides downstream of the PTC define an NMD substrate[5]. An alternative model, "faux 3'UTR", proposes that the distance from the termination codon to the poly (A) tail provides for the targeting of NMD due to the competitive binding of the major NMD regulator, up-frameshift factor 1 (UPF1), and cytoplasmic poly (A) binding protein 1 (PABC1) with eukaryotic polypeptide chain release factor 3 (eRF3), which interacts with eRF1 and recognizes termination codons including the PTC[6]. Recent genome-wide exome and transcriptome approaches, which investigated genetic variants predicted to be targeted by NMD and their transcriptome profiles, discovered that a significant proportion of transcripts predicted as NMD targets supports the EJC model[7,8].

NMD efficiency varies across transcripts (e.g., ~20% in β-globin[9], or Triosephosphate isomerase (TPI)[10], ~4% in T-cell receptor β (TCR-β)[11] or Immunoglobulin (Ig)[12,13]) depending on the position of the PTC[14–16], tissue type[7,8,17], oncogenesis[18], and stress conditions[19–24]. Genome-wide studies also revealed that a significant proportion of transcripts were predicted to trigger NMD escape from degradation by unknown mechanisms[7,25,26]. Conditional NMD inhibition, "NMD escape", seems to be a general feature of this process since a subpopulation of NMD substrates escapes from NMD[27,28]. However, little is known about how and what leads to NMD escape.

Cell variation is a fundamental characteristic in the process of differentiation, oncogenesis, aging, and many other biological processes[29]. The technological advances of single-cell approaches, such as fluorescence-activated cell sorting (FACS), single-cell RNA sequencing, and single-molecule fluorescence in situ hybridization (smFISH) have uncovered dramatic variability and diversity in cell populations where heterogeneity was previously unappreciated (Review[30,31]). The ordinary approaches to determine NMD efficiency, characterizing half-lives of PTC-containing transcripts compared with normal transcripts are generally ensemble assays. These determine the mean of NMD efficiency but are unable to investigate the heterogeneities in the population with single-cell resolution.

## Results

**Application to investigate cell-to-cell variability in NMD.** In this study, we established an assay to assess NMD efficiency in individual cells. We used a bi-directional system, which allowed the simultaneous and calibrated expression of two transcripts from two open reading frames (ORFs) under the same promoter within the same cell[32]. This allowed us to compare genes with or without PTC in the same cell, eliminating extrinsic variability. We confirmed that the ponasterone A (PonA)-inducible bidirectional promoter expressed equally in both directions. We used a construct, which expressed wild-type β-globin (Gl) ORF in either direction but contained different stem-loops in the 3′UTR (MS2 or PP7 stem-loops, Fig. 1a, b; WW), as well as swapping their locations as a control (Fig. S1; Switch). These were transiently transfected in human osteosarcoma U2OS PonA cells. After induction for 24-hr, cells were fixed for smFISH. The individual

mRNA molecules were hybridized using fluorescent dye conjugated probes to their respective stem-loops (Fig. 1c and S1) for fluorescence microscopy. Individual mRNAs were counted using FISH-quant[33], and correlation of expression from either side of the promoter was determined in a scatter plot (Fig. 1d and S1). The linear correlation between the expression of the mRNAs showed an equivalent induction of transcription from either orientation of the PonA bi-directional promoter. We also found similar results after a 1-hr induction (Fig. S2).

The approach was then applied to an NMD target combined with a wild-type sequence. We performed the same experiment with a PTC-containing ORF in one of the orientations (Fig. 1a, e; WP). Individual mRNAs were counted as described below, then each number of transcripts was represented in a scatter plot (Fig. 1f). In contrast to WW transfected cells, we found that WP transfected cells showed a different regression curve with a steeper slope (slope = 1.5, $R$ squared = 0.31, Fig. 1f). This is due to the reduction of PTC-containing mRNA by NMD. Notably, the low R-squared value of the regression indicated variable NMD efficiencies, where the population of cells escaping NMD was apparent (Fig. 1g, (i) and (ii)). This result demonstrated the variability of NMD efficiency, not detectable using ensemble measurements, such as RT-qPCR and northern blotting. The ability to quantify the escaping population allows an analysis of its characteristics. The fraction of NMD escape in cell population was inspected in the density plot of the NMD efficiency in WP expressing cells (Fig. S3). This showed that 13.5% of cells escaped from NMD and 14.4% of cells showed moderate NMD escape. 72.1% of cells exhibited efficient NMD.

One mechanism to explain how some cells escape from NMD could be from the reduction of translational activity since translation is required for NMD. Certain cellular stress conditions such as cellular hypoxia and amino acid deprivation cause translation shut-off through eIF2α phosphorylation and result in inactivation of NMD[21–24]. To test if these cells escaped from NMD due to the inactivation of translation, we performed simultaneous detection of NMD efficiency and translational activity using smFISH and the click-it HPG (L-Homopropargyl-glycine) system. HPG is the amino acid analog of methionine, and the HPG translation detection system is based on the fluorescence labeling by HPG incorporation into nascent proteins. As shown in Fig. S4, treatment by cycloheximide (CHX), an inhibitor of translation elongation, decreased the detection of HPG incorporation (Fig. S4a, b), confirming successful detection of translation activity using this system. The simultaneous detection of NMD efficiency and translation activity showed no correlation (Fig. S4c). This indicated that cells could escape from NMD even when they are actively translating. In addition, NMD efficiency had no correlation with cell size, indicating that the cell cycle phase, the nuclear size or the level of expression were not involved in the escape (Fig. S5).

We also developed a fluorescent NMD reporter system, by fusing EGFP or mCherry coding sequences upstream of either β-globin gene (Fig. 2a, b). The efficiency of NMD can be monitored as a ratio of the two fluorescence intensities in single cells (Fig. 2c). This allowed the isolation of the specific cell population that exhibited NMD escape by fluorescence-activated cell sorting (FACS). Cells expressing mCherry-tagged wild-type β-globin and EGFP-tagged wild-type or EGFP-tagged PTC-containing β-globin constructs were transiently transfected into HEK293T ponA (Fig. 2d, e, f) or U2OS ponA (Fig. 2g, h, i) cells, and transcription was induced by Pon A for 24-hr. The intensities of mCherry and EGFP in the cells were determined using flow cytometric analysis. The intensity of mCherry represents the control; NMD efficiency can be quantified by a ratio of two fluorescence intensities expressed in each cell. Most wild-type cells exhibited both EGFP

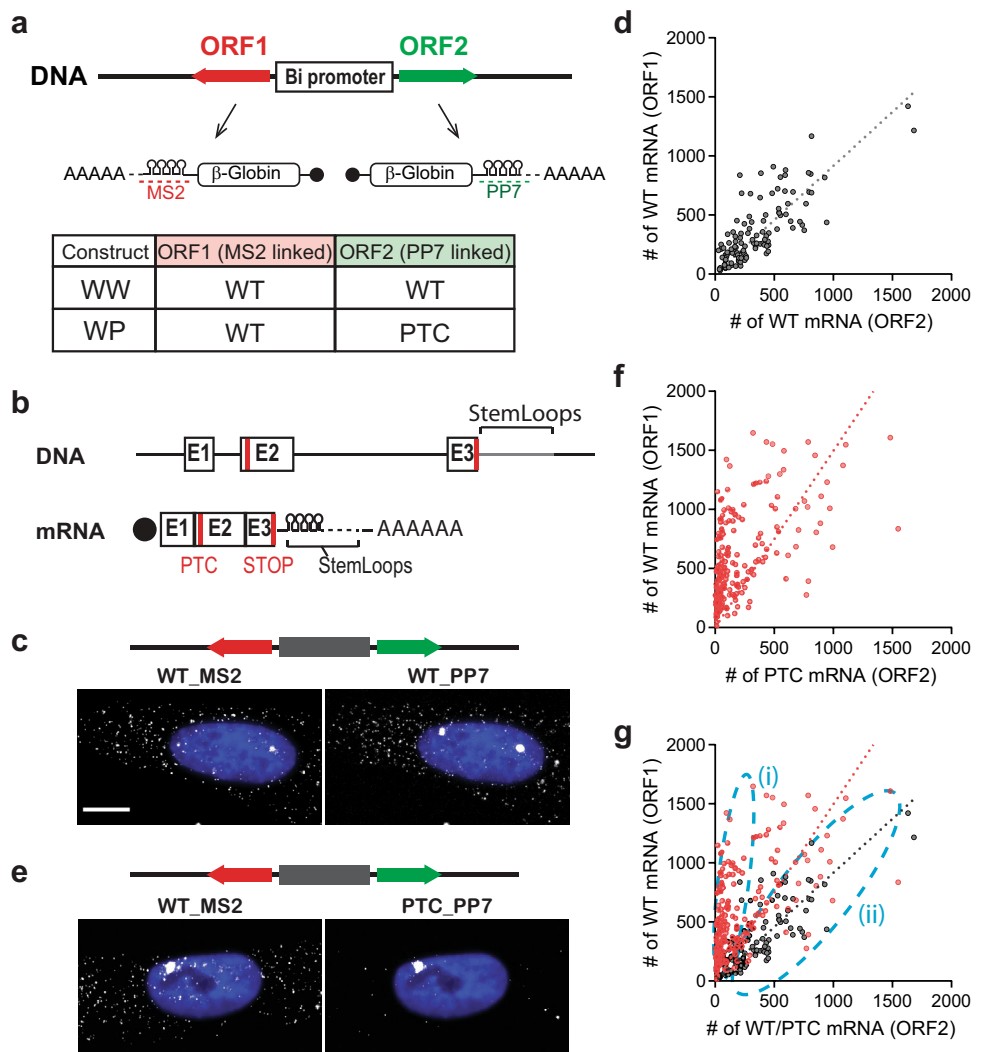

**Fig. 1 Cellular variability of NMD. a** Schematic of PonA inducible bi-directional promoter expressing NMD reporter Gl genes. These transcripts contain MS2 or PP7 sequences in the 3'UTR that hybridize with Quasar 670-labeled (red dotted line) or Quasar 570-labeled (green dotted line) FISH probes. The table shows the mRNAs expressing from the bi-directional promoter from each construct. **b** Structures of NMD reporter Gl (DNA; upper and mRNA; lower) containing a PTC at position 39 (PTC). Horizontal lines represent introns, 5'-UTR and 3'-UTR, and boxes represent each of the three Gl exons (E1-3) joined by splicing-generated exon-exon junctions. Black dot, cap structure; Red lines, termination codons; STOP, normal termination codon; AAAAAA, poly(A) tail. **c**, **e** smFISH images of U2OS PonA cells co-expressing Gl WT (left; expressed from ORF1) and Gl WT (**c**) or PTC (**e**) mRNA (expressed from ORF2) from a bi-directional promoter. NMD reporter was transiently expressed. MS2 mRNAs and PP7 mRNAs were simultaneously labeled with Quasar 670- and Quasar 570- conjugated FISH probes. Note the lack of cytoplasmic mRNA due to the PTC in **e**. The nuclei were stained by DAPI (blue). Bar = 10 um. **d**, **f**, **g** The Scatter plots show the number of Gl WT (ORF1) and Gl WT or PTC (ORF2) mRNAs expressing from WW (**d**, black), WP (**f**, red), and both were superimposed in **g**. $n_{cells}$ = 117 and 140 for WW and WP constructs. Each dot in scatter plots denotes the number of Gl WT (ORF1; y-axis) and WT or PTC (ORF2; x-axis) mRNAs in a single cell. Dotted lines indicate regression lines. Sub-populations of cells with differential NMD efficiency ((i) efficient NMD and (ii) escape from NMD) are shown as light dotted blue lines.

and mCherry double-positive and a scatter plot of the fluorescent intensities represented a linear correlation, confirming the equivalent expression of both proteins when expressed from the bidirectional promoter (Fig. 2d, g, f, i). In contrast, the scatter plot of the fluorescent intensities is skewed toward lower EGFP intensities when EGFP-tagged wild-type β-globin mRNAs was replaced with the NMD substrate, indicating that these mRNAs were degraded. However, some of the cells demonstrated an EGFP/mCherry ratio more characteristic of wild-type which represented the cells undergoing NMD escape (Fig. 2e, h, f, i). A similar pattern was observed in both HEK293T and U2OS cell lines, suggesting NMD escape in this reporter is common to these cell lines. We also established stable cell lines expressing WW or WP (Fig. S6a, b, c), FP-WW, or FP-WP (Fig. S6e, f) constructed from the specific genome loci of Flp-In U2OS PonA cell line

(clone # A33-8[34]) using the Flp-In system (Thermo Fisher Scientific Inc.) since stable cell line expressing NMD reporter might be more homogeneous compared to cells transiently transfected. As expected, we found fewer cells escaped from NMD in Flp-In WP cells detected by smFISH (Fig. S6b, c) as well as by flow cytometry compared to cells transiently transfected (Fig. S6d, e). Interestingly, we found that transient transfection alone also triggered NMD escape in the stably-expressing NMD reporter (Fig. S6f), indicating that cellular stress caused by DNA transfection may induce NMD escape.

**NMD escape is associated with reduction of SMG1.** NMD involves the core regulator up-frameshift proteins (UPFs). UPF1 initiates the NMD process by interacting with eRF3 at the PTC[35].

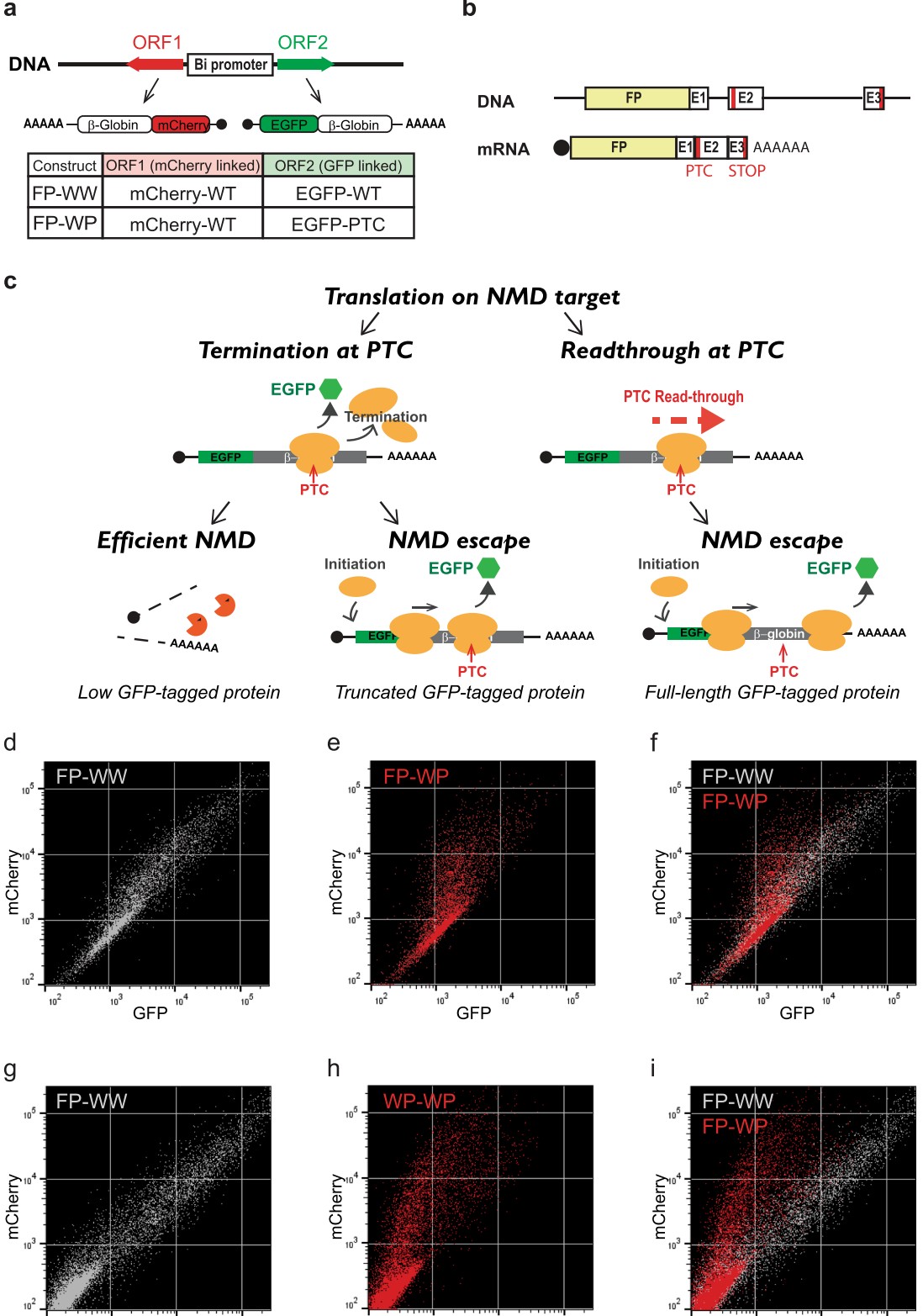

The interaction of PTC-bound UPF1 with EJCs, which consists of UPF2, UPF3A, or UPF3B, detects the PTC in mammals[36–38]. Besides UPFs, NMD involves SMG proteins, such as SMG1, SMG5, SMG7, and SMG6 in mammalian cells. The phosphatidylinositol-kinase related kinase (PIKK) SMG1 phosphorylates UPF1 and activates further degradation of mRNA[35,39]. This degradation is carried out by the endonuclease SMG6[40,41] as well as the heterodimer SMG5-SMG7, which recruits the CCR4-NOT deadenylation complex[42–44] that leads to decapping, deadenylation and exonucleolytic degradation[45,46]. Since NMD escape could result from reduced expression of one of these known NMD regulators in individual cells, we correlated NMD efficiency with protein levels of NMD regulators, UPF1, SMG1, and SMG6, along with the phosphorylation of UPF1, using our

**Fig. 2 Single-cell analysis of NMD using flow cytometry. a** Schematic of translational reporter Gl genes. These transcripts contain EGFP or mCherry coding sequences upstream of the Gl gene. The table shows the mRNAs expressing from the bi-directional promoter from each construct. **b** Structures of Gl DNA (upper) and mRNA (lower) containing a PTC at position 39 in Gl coding region (PTC). Horizontal lines represent introns, 5′-UTR and 3′-UTR, and boxes represent each of the three Gl exons (E1-3) joined by splicing-generated exon-exon junctions. Black dot, cap structure; Red lines, termination codons; STOP, normal termination codon; AAAAAA, poly(A) tail; FP, fluorescent protein. **c** Schematic of EGFP expression from NMD reporter construct and expected results. EGFP expresses from the FP-WP construct either when PTC was subject to readthrough or when the PTC was recognized, generating a truncated GFP-containing protein but failed to trigger decay of the mRNA. Low EGFP expression might be detected even when NMD was successfully triggered since EGFP translates prior to PTC recognition. FP construct (FP-WW; Wild-type and Wild-type or FP-WP; Wild-type and PTC) was transiently transfected into **d**, **e**, **f** HEK293T ponA or **g**, **h**, **i** U2OS ponA cell line. Each dot denotes the intensity of mCherry and EGFP in a FP-WW (Gray, **d**, **g**) or FP-WP (Red, **e**, **h**) expressing single cell and both were superimposed (**f**, **i**). Transcription was induced with 20 nM PonA and the intensities of mCherry and GFP were detected using BD FACSArea II. 20,000 cells were analyzed and single cells were determined using SSC (side scatter) and FSC (forward scatter), and live cells were selected as DAPI negative cells.

imaging approach. The relevant construct was transiently transfected into U2OS ponA cells and induced for 24-hr. UPF1, UPF1 phosphorylation, SMG1, or SMG6 were labeled by immunofluorescence using their specific antibodies followed by anti-rabbit Alexa Fluor 647. The fluorescence levels of EGFP, Cherry, and IF labeling NMD regulators were detected by fluorescence microscopy and their intensities in each cell were determined using CellProfiler[47–49]. The results showed that NMD efficiency was correlated with the level of SMG1 (Fig. 3a, b, c); less correlation was found with the level of UPF1 or with SMG6 (Fig. S7). A correlation of NMD efficiency (a negative correlation with NMD escape) with phosphorylation of UPF1 was also evident (Fig. 3d, e, f). This is likely due to the reduction of SMG1 which phosphorylates UPF1. The negative correlation of NMD efficiency with the level of SMG1 is consistent with previous work showing the overexpression or downregulation of SMG1 increases or decreases NMD efficiency respectively[39,50].

SMG1 is a multitasking player in the maintenance of telomeres, the regulation of apoptosis and several cellular stress responses including DNA damage, oxidative and hypoxic stresses[51–55]. SMG1 exhibits some functional overlap with another PIKK family member, ataxia-telangiectasia mutated (ATM) protein. Kinase activity in both SMG1 and ATM are stimulated in response to genotoxic stress and phosphorylate their downstream target p53 on serine 15, to coordinate downstream stress-induced signaling pathways[52,56,57]. To address if NMD escape is induced by the DNA stress response, we detected ATM autophosphorylation at Ser 1981 (ATM S1981P) as a DNA double-strand break (DSB) marker. Although, the level of ATM did not correlate with NMD efficiency (Fig. S8a, b), the co-detection of ATM S1981P by IF using a specific antibody with NMD fluorescence reporter revealed that subcellular localization of ATM S1981P changed from punctate bright spots (Fig. S8c (a)) to a homogeneous distribution (Fig. S8c (b)) in the nucleus when cells escaped from NMD. This uniform distribution pattern of ATM S1981P in the nucleus was consistent with the early phase of ATM activation under DSBs[58]. Although it is not clear the mechanism of the subcellular localization of ATM S1981P, this suggested a link between NMD efficiency and DNA damage stress response.

To test the possibility that SMG1 reduction led to NMD escape in the DNA damage response, the level of SMG1 was detected using a drug, which induces DNA damage such as doxorubicin and etoposide (Fig. 4 and Fig. S9). IF against anti-SMG1 antibody was performed on cells treated with doxorubicin or etoposide for 24-hr. The level of SMG1 was significantly reduced with treatment of doxorubicin (Fig. 4a, b) and exhibited recovery when cells were washed out after a 4-h doxorubicin treatment (4-hr pulse). In contrast, the level of SMG1 exhibited no or mild reduction when cells were treated mid- or high- concentrations of etoposide (10 μM or 50 μM) (Fig. S9a, b). This could be due to the difference in drug actions[59]. Despite the mild reduction of SMG1,

we detected an increase in the cell fraction with NMD escape under high concentrations of etoposide using flow cytometry (Fig. S9c, d). While we were unable to measure NMD escape using our single-cell approach due to the high fluorescent background of doxorubicin[60], we detected the significant NMD inhibition in doxorubicin-treated cells using RT-qPCR.

**NMD escape occured with or without PTC readthrough.** Since PTC recognition is a critical requirement for NMD, translation readthrough might be a key mechanism of NMD escape. Translational readthrough happens due to failure to terminate translation and inserts a near-cognate tRNA for the termination codon that allows elongation to continue. This readthrough at a PTC presumably decreases the likelihood of the mRNA to undergo NMD, also in following rounds of translation of the mRNA since translational readthrough removes the exon junction complexes (EJCs) downstream of the PTC[61] required for most of NMD activation[5]. NMD escape also may occur due to the failure of the mRNA to degrade after PTC recognition. Therefore, NMD escape may take place in two ways: cells fail to trigger NMD because (1) the ribosome fails to recognize the PTC and reads through it, or (2) translation terminates at a PTC but NMD regulators fail to trigger decay. Since the PTC is located downstream of the fluorescence coding sequence in the NMD fluorescence reporter system where the translation of EGFP occurred prior to PTC recognition, the EGFP intensity could not determine whether NMD escape resulted from the failure of PTC recognition itself or following degradation after successful PTC recognition. The molecular weight of the EGFP-tagged protein in the western blotting analysis determined whether the PTC was readthrough or successfully recognized resulting in the production of full-length or truncated EGFP-tagged proteins (Fig. 5). The constructs were transiently transfected into HEK293T ponA cells and induced for 24-hr. The specific cell population that escaped from NMD (higher EGFP intensity) or exhibited efficient NMD (lower EGFP intensity) were isolated using FACS (Fig. 5a). The protein levels and sizes of EGFP and mCherry in each cell population were determined using western blotting (Fig. 5b). Since PTC containing EGFP-globin mRNA was degraded efficiently by NMD, a band representing EGFP was not detectable in cells with higher NMD efficiency. In contrast, two distinct molecular sizes of EGFP bands were detected in the cells exhibiting NMD escape. The band with higher molecular weight represented full-length EGFP-globin protein, that resulted from translational readthrough as the identical size of the band was found in the cells expressing EGFP-tagged wild-type β-globin proteins. Interestingly the band with smaller molecular weight was detected in the cells showing NMD escape, which represented the truncated EGFP-tagged β-globin proteins resulting from the PTC. The relative intensity of each band indicated that almost 80% of translating ribosomes successfully terminated at the PTC,

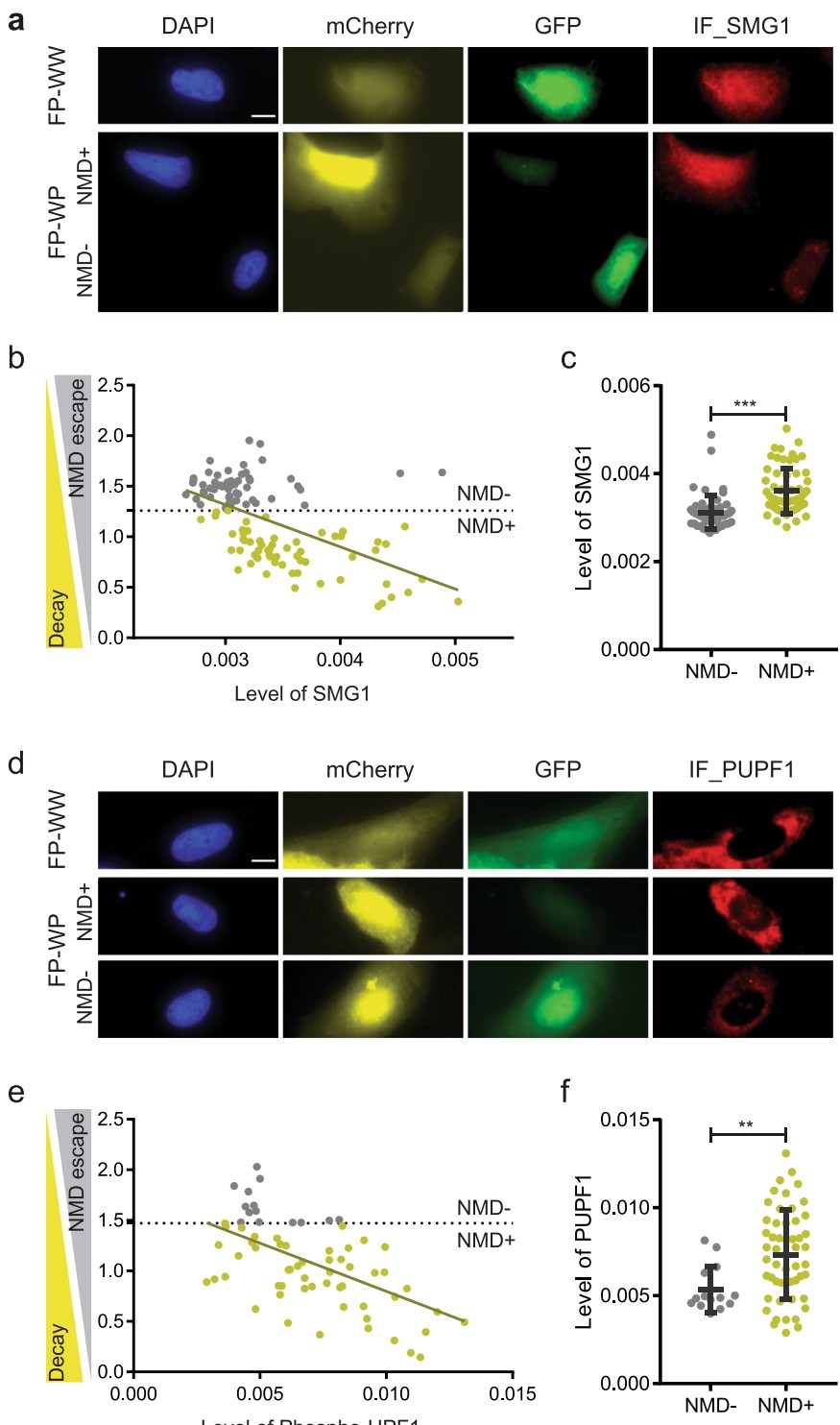

and 20% engaged in translational readthrough during NMD escape.

**Bi-directional NMD reporter indicated cellular NMD efficiency.** In addition to assessing the protein levels of NMD regulators and NMD efficiency (Fig. 3), we also determined the mRNA levels of NMD regulators by RT-qPCR. The constructs were transiently transfected into HEK293T ponA cells or U2OS ponA cells and the FACS isolated cell populations with the representative NMD efficiencies (Fig. 5a) were subjected to quantitative RT-qPCR analysis. In contrast with the lower protein

levels of SMG1 in the cells that have escaped (Fig. 3b, c), the relative mRNA enrichment of NMD regulators compared with the cells with efficient NMD showed that mRNA levels of SMG1 was higher in both HEK293T ponA and U2OS ponA cells (Fig. 5c, d). The inconsistency between lower protein (Fig. 3) and increased mRNA level of SMG1 in the NMD escape cells could be explained by the known auto-regulatory feedback of the NMD mechanism[62,63]: NMD targets mRNAs of its own regulators, most of which contain longer 3′UTRs. The preceding reduction of SMG1 protein might trigger the NMD inhibition, therefore, mRNA of NMD regulators might be stabilized in NMD escape

**Fig. 3 Correlation of NMD efficiency with NMD regulator SMG1 and phosphorylation of UPF1.** Simultaneous detection of fluorescence NMD reporter and SMG1, or phospho-UPF1 (PUPF1). **a**, **d** Immunofluorescence Images of SMG1 or PUPF1 in U2OS cells expressing NMD fluorescence reporter. FP construct (FP-WW; Wild-type and Wild-type or FP-WP; Wild-type and PTC) was transiently transfected into U2OS ponA cell line. Transcription was induced with 20 nM PonA for 24-hr. Bar = 10 μm. SMG1 or PUPF1 was detected by immunofluorescence using anti-SMG1 or anti-phospho-UPF1 antibody followed by Alexa Fluor 647 linked anti-rabbit IgG. The intensity of mCherry, EGFP, and the level of SMG1 or PUPF1 (IF_SMG1 or PUPF1), DAPI-stained nuclei were detected using fluorescence microscopy. **b** Correlation of NMD efficiency with the level of SMG1 or **e** PUPF1. X- or y- axis shows mean intensity of IF or NMD efficiency. The NMD efficiency in single cells was calculated by the intensity of EGFP normalized by the intensity of mCherry (EGFP/mCherry ratio). Single dots denote fluorescence intensity of IF and normalized NMD efficiency from single cells. The level of NMD escape (NMD +; higher NMD efficiency, NMD −; NMD escape) was defined by the 10 percentile of GFP/mCherry ratio in FP-WW expressing cells (horizon dotted line in b&e). No correlation between EGFP/mCherry ratio and the level of SMG1 or PUPF1 was found in FP-WW expressing cells. The line indicates the regression line. **c** The level of SMG1 or **f** PUPF in FP-WP expressing cells are shown with NMD efficiencies. Mean intensity of EGFP, mCherry and IF representing the level of SMG1 or PUPF1 in individual cells were determined using Cellprofiler software[47–49]. $n_{cells}$ = 116 (60 and 56 in NMD + and NMD −) and 68 (53 and 14 in NMD + and NMD −) for SMG1 and phosphor-UPF1 IF. $P$ values were determined using two-tailed unpaired t-tests (***$P$ < 0.0001, **$P$ = 0.0067). Error bars = Standard deviation in cell populations. The statistical analysis was performed by Graphpad prism software.

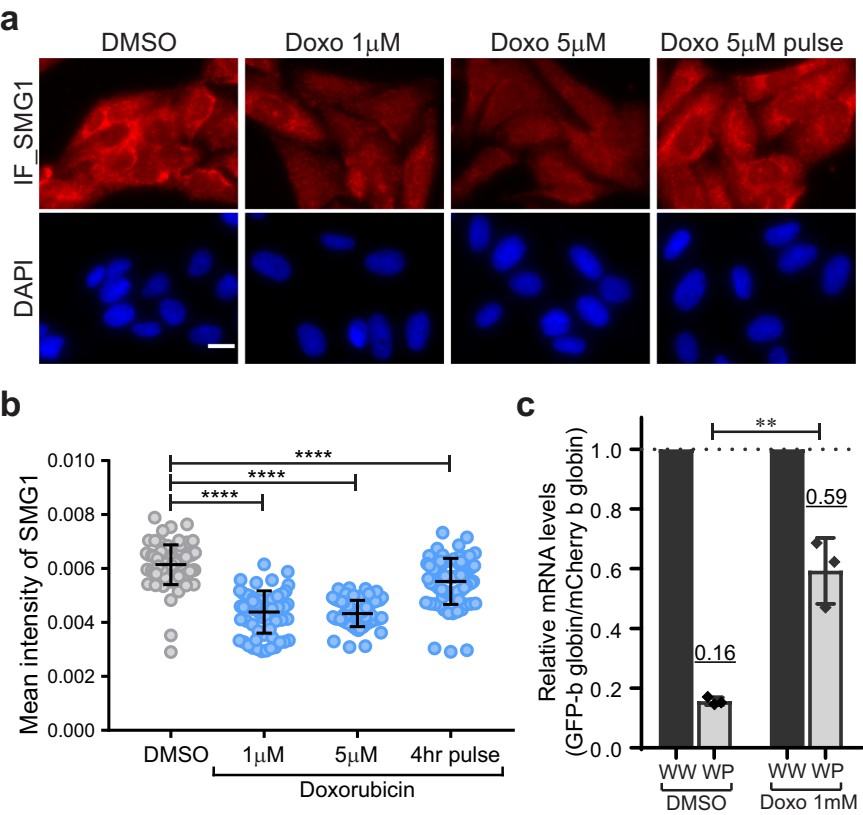

**Fig. 4 SMG1 is decreased by a DNA damage inducer. a** SMG1 detection in U2OS cells by immunofluorescence (IF) under the treatment of doxorubicin (Doxo). Cells were treated with doxorubicin for 24-hr with the indicated concentrations. Doxorubicin treatment was removed after 4-hr and transcription was induced with 20 nM PonA in the cells (Doxo 5 μM pulse). SMG1 was detected by IF using anti-SMG1 antibody followed by Alexa Fluor 647 linked anti-rabbit IgG. Bar = 20 μm. **b** The mean intensity of SMG1 in cytoplasm was detected in **a**. $n_{cells}$ = 82, 67, 71, 78 for DMSO, 1 uM, 5 uM, or 4-hr pulse doxorubicin treatment. **b** Single dots denote fluorescence intensity of SMG1 intensity in individual cells. The mean intensity of cytoplasmic SMG1 in individual cells were determined using Cellprofiler software[47–49]. $P$ values were determined using two-tailed unpaired t-tests (****$P$ < 0.0001). Error bars = Standard deviation in cell populations. **c** NMD efficiency was determined by the quantitative mRNA detection of NMD reporter. Relative EGFP-β globin mRNA levels (Wild-type or PTC-containing β globin in WW or WP expressing cells) were determined by RT-qPCR. mCherry-β globin mRNA (wild-type β globin mRNA in each WW or WP expressing cells) was used as a control. Relative GFP-β globin mRNA levels under high concentration of doxorubicin (5 μM) was unable to detected due to the significant reduction of reporter expression. The underlined number above the data points indicates the mean value. $P$ values were determined using two-tailed unpaired $t$ tests (**$P$ = 0.0025). Error bars = Standard deviation from three independent experiments. The statistical analysis was performed by Graphpad prism software.

cells. The mRNA levels of UPF2, UPF3A, and SMG6 did not correlate with NMD efficiency in both cell types. The differential correlation of UPF3B and SMG7 mRNA for the two cell types could be explained by cell-type-specific auto-regulatory feedback as reported previously[62]. Moreover, NMD efficiency of

endogenous NMD-targets in each FACS-isolated cell population was assessed as a control since NMD escape may be transcript-dependent (or reporter-dependent). Using RT-qPCR analysis, we detected higher mRNA levels of endogenous NMD targets, except ANTXR1 mRNA, in HEK293T ponA (Fig. 5e) and U2OS ponA

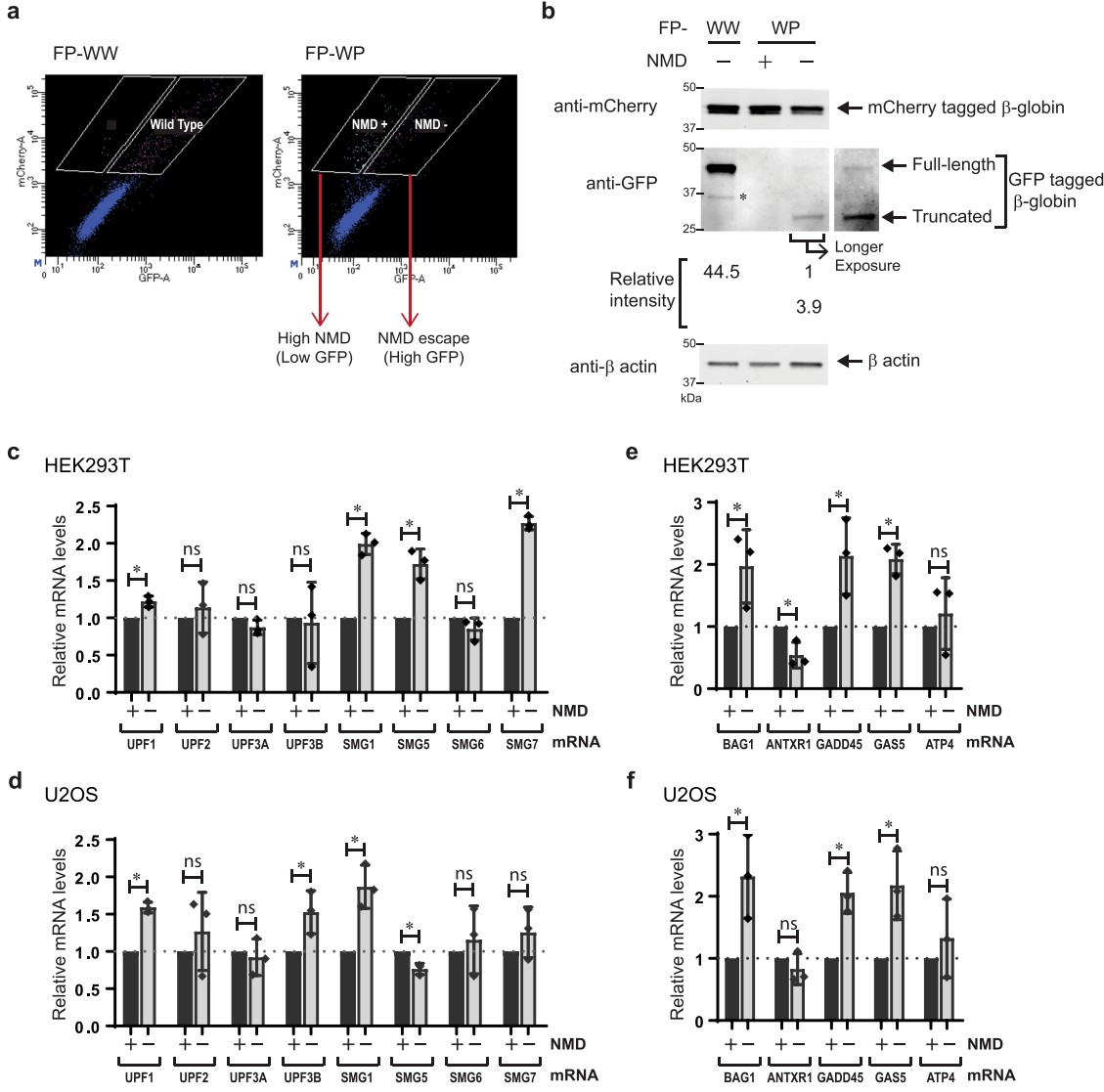

**Fig. 5 Cell isolation correlated with NMD efficiency. a** Flow cytometric analysis of FP-WW or FP-WP expressing HEK293T PonA cells. Based on the mCherry and EGFP intensities in FP-WW expressing cell, the cell populations with distinct NMD efficiency (NMD + ; higher NMD efficiency, NMD −; NMD escape) were determined and sorted using FACS. White boxes indicate the gate for cell isolation for FACS. Single dots denote fluorescence intensities from single cells. **b** The EGFP and mCherry proteins expressed in isolated cell populations in **a** were detected in western blotting. The same number of cells isolated were loaded in each lane. EGFP-tagged wild-type β globin was detected in the first lane (FP-WW). In contrast, no bands of EGFP-tagged β globin was detected in the second lane (efficient NMD cell population (NMD + ) from FP-WP expressing cells). The differential intensities of EGFP bands between lower (truncated EGFP-tagged β globin) and higher (full-length EGFP-tagged β globin) indicated the frequencies of translation termination at PTC and readthrough. Relative intensities of EGFP bands are indicated under the blot detected with anti-GFP antibody. *Degradation product. β actin was detected as a loading control. **c**, **d** Quantitative mRNA detection of NMD regulators in NMD escape cells. Relative mRNA levels of NMD regulators in the cell population with NMD escape (NMD−) compared to the cell population with efficient NMD (NMD + ) in HEK293T ponA (**c**) or U2OS ponA cells (**d**) were determined by RT-qPCR using primers specific for each transcript as indicated above each bar. P values were determined using two-tailed unpaired t-tests (**c**; p* = 0.0068 (UPF1), 0.5216 (ns, UPF2), 0.0867 (ns, UPF3A), 0.8421 (ns, UPF3B), 0.0003 (SMG1), 0.0033 (SMG5), 0.1523 (ns, SMG6), < 0.0001 (SMG7)), **d**; p = 0.0001 (UPF1), 0.4240 (ns, UPF2), 0.6138 (ns, UPF3A), 0.0337 (UPF3B), 0.0067 (SMG1), 0.0054 (SMG5), 0.5841(ns, SMG6), 0.2609(ns, SMG7), ns = not significant). Error bars = Standard deviation from three independent experiments. **e**, **f** Relative mRNA levels of endogenous NMD targets in the cell population with NMD escape (NMD−) compared to the cell population with efficient NMD (NMD + ) in HEK293T ponA (**e**) or U2OS ponA cells (**f**) were determined by RT-qPCR using primers specific for each transcript as indicated above each bar (BAG1; BCL2-associated athanogene 1[37,76], ANTXR1; ADP Ribosylation Factor Related Protein 1[77], GADD45; Anthrax toxin receptor 1[77], GAS5; growth arrest specific 5[77], ATF4; activating transcription factor-4[4]. β actin (ACTB) was used as a control. P values were determined using two-tailed unpaired t tests (**e**; p* = 0.0464 (BAG1), 0.0186 (ANTXR1), 0.0353 (GADD45), 0.0014 (GAS5), 0.5634 (ns, ATP4), **f**; p = 0.0274 (BAG1), 0.2974 (ns, ANTXR1), 0.0052 (GADD45), 0.0210 (GAS5), 0.4266 (ns, ATP4), ns = not significant). Error bars = Standard deviation from three independent experiments.

(Fig. 5f) cell lines that escape compared to the cells with efficient NMD. This suggests that NMD escape has a general effect on the transcriptome.

## Discussion

The intercellular variability in cell populations has been acknowledged by advanced technology such as single-RNA seq, however, general biochemical assays addressing post-transcriptional regulation are not capable of single-cell detection. In this study, we established single-cell detection of NMD efficiency using a NMD reporter construct containing two ORFs, which express wild-type and PTC-containing mRNA bi-directionally in the same cells. The expression of wild-type mRNA in the same cells verifies the proper biological activity representing an internal control in individual cells, thus it minimizes the variables related to gene expression other than NMD. Using this reporter system, single mRNA detection with smFISH revealed a variable range of NMD efficiencies in the cell population, providing evidence of cell-to-cell heterogeneity of NMD. In addition, we established a dual-fluorescence NMD reporter that was compatible with the flow cytometric detection and cell isolation using FACS. Our dual-fluorescence NMD reporter system provided a conventional single-cell detection of NMD, which could be accomplished by simple fluorescence detection without requiring RNA purification or fluorescence labeling. Moreover, cell isolation with individual NMD efficiencies by FACS enabled the investigation of rare but informative cells that were not detectable in the general biochemical approach.

NMD escape appears to be a general biological characteristic for most NMD targets. Previous studies have also supported that conditional NMD inhibition is a part of post-transcriptional regulation in response to several environmental stimuli or physiological changes. Global translational inhibition is known to be a mechanism of NMD inhibition under several cellular stress conditions[51–55]. However, we characterized NMD escape while cells were undergoing active translation, suggesting the mechanism that alters the NMD efficiency did not require translation inhibition.

Comparison of NMD efficiency using the dual-fluorescence NMD reporter with the cellular level of proteins using IF for known NMD regulators identified a correlation with SMG1 levels, which plays a key role in DNA damage response. We also showed that the pharmacological treatment inducing DNA damage reduced the level of SMG1 and inhibited NMD, suggesting a crucial regulation of DNA damage response by NMD. Although only a mild reduction was observed in etoposide-treated cells, we found a significant reduction of SMG1 with the treatment of doxorubicin, which led to NMD inhibition. Notably, our result of the doxorubicin-induced NMD inhibition was consistent with a previous study, reporting that the doxorubicin-induced UPF1 cleavage during apoptosis pathway[64]. It is unclear how the reduction of SMG1 and the cleavage of UPF1 are linked in NMD inhibitory mechanism. Since the DNA damage response is tightly associated with the apoptosis pathway, further investigation needs to be addressed to determine the detail of NMD inhibitory mechanism under treatment of doxorubicin, and the role of SMG1 in the DNA damage response.

Since PTC recognition is a key event to trigger efficient NMD, we also investigated whether cells escape from NMD with PTC recognition or readthrough. Various translational readthrough detection assays have been developed in previous studies. The most common approaches are dual reporter systems using either two fluorescent proteins, Renilla, and Firefly luciferases that are tandemly linked via stop codon in the same transcript[65–67]. These assays are useful to determine the inter-molecule readthrough

efficiency at PTC but most of them were not optimized for the detection of NMD and were ensemble measurements not capable of single-cell detection. Interestingly, isolation of NMD escape cells followed by western blotting determined that 20% of translating ribosomes read through the PTC in the cell population that escapeed surveillance even though translational readthrough is generally known as a rare event (<1%)[68]. Notably, we detected this higher level of readthrough protein products only in the NMD-escape cell population but not in the efficient NMD cell population. This could be explained by the improved sensitivity in the selective NMD-escape cell population in our approach. This represented the advantage of our single-cell approach, which provides a sensitive tool to investigate a minor cell population that could not be detected previously. Since the efficiency of translational readthrough is altered by the type of termination codons (UAA, UAG, or UGA)[69] as well as the sequence surrounding the termination codon[70–72], the efficiency of NMD in each NMD transcript might be tightly associated with the translational read through events. Our system enableed the detection of translational readthrough and successful PTC recognition in NMD escape cells, and provideed a sensitive assessment to elucidate the pathways whereby the PTC was recognized or not.

A study regarding NMD escape potentially provides an approach for drug discovery targeting of nonsense suppression. Inducing translational readthrough at the PTC is an attractive strategy for nonsense-associated disease-causing genes. The major limitation of nonsense suppression therapy is a significant reduction of nonsense transcripts by NMD. Our approach offers a sensitive assay for translational readthrough that will provide valuable insights into unknown regulatory pathways resulting in NMD escape. Finally, this approach can potentially expand into future studies that investigate the role of NMD in many biological phenomena such as cell differentiation, memory consolidation, virus infection, tumorigenesis, and immune response[19,20].

## Methods

**Cell Lines and Tissue Culture**. Human U2OS PonA or Flp-In U2OS PonA cell line was generated from U2OS cells (ATCC, HTB-96™) or Flp-In U2OS cells (gift from Dr. Senecal[34]) by stable transfection of pERV3, which expresses synthetic VP16-glucocorticoid/ecdysone receptor (VgEcR) and retinoid-X-receptor (RXR) that are required for induction for the transcription of PonA promoter as described previous study[73]. Human embryonic kidney 293 T PonA cell was generated from HEK 293 T cell (ATCC, CRL-3216™) by stable transfection of pERV3-zeocin vector, which is replaced neomycin drug selection gene in pERV3 to Zeocin gene due to the neomycin resistance of HEK293T cells. Both cell lines were grown at 37 °C and 5% $CO_2$ in Dulbecco's Modified Eagle's Medium (DMEM) supplemented with 4.5 g/l of glucose, 10% FBS and 1% penicillin-streptomycin.

**Cell transfections, single-molecule fluorescence in situ hybridization (smFISH), immunofluorescence (IF) and fixed cell image acquisition**. U2OS PonA cells were transiently transfected with Bi-directional PonA construct, pFRT-PonA-BI-Gl-WT-24xMS2-WT-18xPP7 (WW), pFRT-PonA-BI-Gl-WT-24xMS2-PTC-18xPP7 (WP), or PonA-BI-Gl-WT-18xPP7-WT-24xMS2 (Switch) with the Amaxa Nucleofector system, then spread onto collagen-coated coverslips. One day after transfection, transcription was induced by supplement with 20 nM PonA for 24-hr (steady-state) in Fig. 1 and S1, and 1-hr for Supplemental Fig. S2. The cells were fixed in 4% paraformaldehyde for 15-min at room temperature, then permeabilized with 0.5% Tween in PBS for 15-min at room temperature. smFISH was performed as previously described[74]. In Figs. 2, 3, and 5, U2OS or HEK293T PonA cells were transiently transfected with pFRT-PonA-BI-Gl-GFP-WT-mCherry-WT (FP-WW) or pFRT-PonA-BI-Gl-GFP-WT-mCherry-PTC (FP-WP). For Fig. 4 and S6 (FlpIn), U2OS FRT PonA cells that were stably transfected with WW, WP, FP-WW or FP-WP. Transcription was induced by supplement with 20 nM PonA for 24-hr (steady-state). For IF, the cells were fixed in 4% paraformaldehyde for 15-min at room temperature, then permeabilized with 0.5% Tween in PBS containing 3% BSA for 15-min at room temperature. The cells were incubated with anti-UPF1 (1/1000 dilution, gift from Dr. Maquat), anti-phospho UPF1 (1/200 dilution, Millipore, 07-1016), anti-SMG1 (1/200 dilution, Cell Signaling 9149 S) or anti-SMG6 (1/200 dilution, invitrogen PA5-60165), anti-ATM (1/200 dilution, Novus Biologicals, NB100-104SS), anti-ATM S1981P (1/500 dilution, Active Motif

#39530) for 1-hr at room temperature. After 10-min washing with PBS for three time at room temperature, the cells were incubated with Alexa Fluor 647 secondary anti-rabbit IgG antibody (Thermo Fisher Scientific, cat# A21245), or anti-mouse IgG antibody (Thermo Fisher Scientific, cat# A21236) at a concentration of 2 µg/mL in PBS for 1-hr. After 10-min washing with PBS for three time at room temperature, the cells were mounted with Prolong Diamond Antifade Mountant with DAPI (Thermo Fisher Scientific. P36966). Images were acquired with an Olympus BX61 widefield, epifluorescent microscope using a 60×1.4 PlanApo objective. Filter sets were used for DAPI (Semrock model DAPI-5060C-Zero), Cy3 (Chroma model 41007), Cy3.5 (Chroma model SP103v1), and Cy5 (Semrock model Cy5-4040C-Zero), with an EXFO X-Cite Series 120 PC metal halide light source, Photometrics Cool SNAP HQ CCD camera, Olympus Type-F immersion oil (nd 1.516) and Molecular Devices Metamorph acquisition software (7.8.10.0.) Cells were optically sectioned using a 0.25 µm Z step, spanning a 10.0 µm Z depth in total. Exposure times of 20 to 500-ms were typically used to acquire each plane in the Cy3, Cy3.5, and Cy5 channels, and ~12-ms were used to acquire each plane in the DAPI channel. All the probes used in smFISH in this study are provided in Supplementary Table 1.

**Detection of translational activity using HPG labeling**.
L-Homopropargylglycine (HPG) labeling was performed as describe in manufacturer's instructions. Briefly, U2OS PonA cells were transiently transfected with Bi-directional PonA construct PonA-BI-Gl WT-24xMS2 PTC-18xPP7 (WP) with the Amaxa Nucleofector system, then spread onto collagen-coated coverslips. One day after transfection, 25 µM HPG (from Click Chemistry Tools, Product No.1067) was added in the methionine-free DMEM medium (Life Technologies Corporation, Part Number 2101302) with or without cycloheximide (50 µg/ml) treatment for 2-hr after transcription induction by supplement with 20 nM PonA for 2-hr. The cells were fixed in 4% paraformaldehyde for 15-min at room temperature, then permeabilized with 0.5% Tween in PBS for 15-min at room temperature. HPG was fluorescently labeled using click-it technology (Click-&-Go Plus 488 Labeling Kit, Product No.1314 from Click Chemistry Tools), then smFISH was performed as previously described[74].

**RT-qPCR**. Total cell RNA extraction, DNase I treatment, and reverse transcription were performed as described previously[61]. Amplifications were performed using Power SYBR® Green PCR Master mix (Applied biosystems, REF 4367659). Primers used in this study are listed in Supplementary Table 2. All samples were analyzed in three times, and the levels of mRNAs were normalized to the level of β-actin mRNA. Relative mRNA levels were determined from CT values according to the ΔΔCT method (Applied Biosystems).

**Flow cytometry analysis and cell sorting**. Flow cytometry detection was performed using cell analyzer LSRII (BD Biosciences) and fcs files were analyzed by FlowJo (BD Biosciences). In Fig. 5, specific cell populations were gated and isolated using FACS Aria II (BD Biosciences).

**Western blotting**. The cells isolated by FACS were lysed using hypotonic lysis buffer[75] and cell lysate was electrophoresed using NuPAGE system (4–12%, Thermo Fisher Scientific), then transferred to nitrocellulose membrane using iBlot 2 (Thermo Fisher Scientific). The membrane was probed with anti-mCherry (1/1000 dilution, Novus Biologicals, NBP2-25157SS), anti-GFP (1/500 dilution, Roche, REF 11814460001), or anti-β actin (Sigma, A1978) antibody following anti-rabbit IRDye® Secondary Antibodies, RDye® 680RD Donkey anti-Mouse IgG Secondary Antibody (1/10,000 dilution, LI-COR, 926-68072), or IRDye® 800CW Goat anti-Rabbit IgG Secondary Antibody (1/10,000 dilution, LI-COR, 926-32211). The signal was detected using Odyssey imaging system (LI-COR Biosciences).

**Data analysis**. The number of mRNA spots were detected and counted in the smFISH images using FISH-quantv3[33]. Standard deviation, two-tailed paired t tests, pearson and R squared were calculated with GraphPad Prism (version 7.04). The levels of HPG and NMD regulators labeled with IF, which was detected fluorescent microscope, were measured based on the area of individual cells determined by CellProfiler3.1.9.

**Statistics and reproducibility**. Each experiment was repeated independently at least three times with similar results.

**Reporting summary**. Further information on research design is available in the Nature Research Reporting Summary linked to this article.

## Data availability
The data supporting the findings of this study are available from the corresponding authors upon reasonable request. Source data for the figures and supplementary figures are provided as a Source Data file. Source data are provided with this paper.

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

## Acknowledgements

This work was supported by NIH grant R01 NS083085 (R.H.S). We thank members of the Singer laboratories for discussions, the Einstein FACS and Genomics cores. We thank A. Senecal for U2OS Flp-In cell line. We also thank L. Maquat for anti-UPF1 antibody. The authors declare no competing financial interests.

## Author contributions

Conceptualization, H.S.; Methodology, H.S.; Formal Analysis, H.S.; Investigation, H.S., R.H.S.; Resources, R.H.S.; Data Curation, H.S.; Writing—Original Draft, H.S. Writing—Review & Editing, R.H.S. Visualization, H.S. Supervision, R.H.S. Funding Acquisition R.H.S.

## Competing interests

The authors declare no competing interests.
