## [Peer Review File · Nature Communications]

Title: Cellular variability of nonsense-mediated mRNA decayREVIEWER COMMENTS

Reviewer #1 (Remarks to the Author):

Sato and Singer use a single-cell approach with a bi-directional reporter to demonstrate an important concept for the RNA field: dramatic cell-to-cell variability of NMD efficiency. With regard to mechanism, they used a fluorescent reporter system coupled with FACS to reveal that cells that undergo NMD escape do this via translational readthrough of the premature termination codon (PTC) or via failure to trigger the mRNA degradation mechanism after successful translation termination at the PTC. While they did not define the precise underlying mechanism responsible, they did show that NMD escape correlates with low levels of the NMD kinase, SMG1, and phosphorylated UPF1. Overall, the manuscript is interesting and the results are convincing. That said, this manuscript is weak with regard to depth, and there are some other concerns:

1. Depth. The authors make various tantalizing discoveries but go no further with them. It is critical that they go more in depth with just ONE of these discoveries. For example, the authors found a correlation between ATM autophosphorylation and NMD escape (Fig. 4A), which led them to speculate that NMD efficiency is linked with DNA damage responses. The authors could examine this DNA damage hypothesis in more detail. Although causal mechanism would be preferred, they could at least perform a correlative study examining other DNA-damage-associated molecules for their association with NMD-escape. A second avenue they could go down is to investigate molecularly what might distinguish NMD-escape vs. NMD-active cells. For example, the authors could do RNA-seq analysis to investigate—at least at the correlative level—what transcripts are associated with these two phenotypes. A third discovery the authors could follow up on is how cells escape NMD despite apparently terminating translation at the PTC (80% of translating ribosomes). The authors may have other ideas on how to enhance depth; the point is to go deeper on one sub-topic to bring this paper to the level of this journal.

2. The following sentence is vague: "The interplay between SMG1 andataxia-telangiectasia mutated (ATM) protein..."(in the last paragraph in page 3) . Are the authors stating there is an interplay between these molecules? If so, what specifically is the nature of this interplay?

3. Another sentence in the same paragraph is also vague: "NMD efficiencies may also correlate with ATM autophosphorylation at Ser 1981." Do these 2 things correlate or not? If so, please give a reference? If no previous evidence, state as hypothesis or rephrase.

4. Rephrase the sentence - "However, the molecular weight of the EGFP-tagged protein and the intensity of band detected..." (second paragraph on page 4) - by omitting "However" and stating as an experiment. Combine this paragraph with the next one, as they are on the same topic.

5. In the first paragraph of the Discussion, the following is claimed: "...thus it provides the precise

detection of NMD efficiency in every cell." This is an overstatement given the considerable evidence that NMD has different branches coupled with the fact that there is considerable variability as to whether a given NMD target mRNA is degraded by NMD in a given cellular context. Thus, it is unlikely that the beta-globin system used by authors will detect "all NMD." This caveat should be explicitly stated.

6. The Discussion is highly redundant with the Results. Some reiteration, such as in the first paragraph, is ok, but after that, the focus should be on implications and speculation, not reiteration of the results.

7. Fig. 1E. Should be changed to 1D so that it is in chronological order in the text.

8. Fig. 3A. The difference in the IF_SMG1 signal between NMD+ and NMD- cells is not obvious. While the data analysis in panel B demonstrates a convincing modest correlation, more cells should be shown (or some other approach should be used) to visually demonstrate the decrease in SMG1 protein expression in NMD escape cells in panel A.

9. Consistency in the figures. It is strongly suggested to show significance r and P values in the "correlation" figures themselves (Fig. 3B/C, 4B), as done in Supplemental Figure S5, not in the legend. To reduce space put the r and P values in the upper portion of the figure rather than to the right..

Reviewer #2 (Remarks to the Author):

Report on manuscript "Cellular variability of nonsense-mediated mRNA decay", by Hanae Sato and Robert H. Singer.

This manuscript describes the development of a bicolor NMD reporter, based on a bidirectional promoter transcribing both a control and a test NMD RNA. By transiently transfecting human cell lines and measuring mRNA levels in single cells by smFISH, the authors show that NMD efficiency varies quite significantly from cell to cell, with some cells showing little NMD activity. The authors confirm this using GFP/mCherry protein reporters, in the same bidirectional vector, and show that low NMD cells correlated with high SMG1 or phospho-UPF1 protein levels, and also high phospho-ATM. Finally, the authors show that low NMD is due to both PTC readthrough and inefficient mRNA degradation, and that low NMD cells, selected by FACS with the two-color reporter, display widespread NMD defects in endogenous NMD target.

The manuscript is clear and straightforward, and the data are clear cut. The observations are interesting and report on an important but previously overlooked property of NMD. This study will certainly form the basis of many more of the kind to come. I have only few comments and I support publication of the paper provided they are addressed.

1-All the experiments are done with transient transfections and this requires quite harsh cell treatments possibly generating cell-to-cell heterogeneity (not all cells are transfected). I thus think that it is important that the authors repeat a couple of their key findings using a stable cell line. I am thinking to Figure 1 of course, but also Figure 5.

2-In the experiment with phospho-ATM (Figure 4), I am not sure I get the point with the diffuse staining. Could the authors elaborate on what causes this diffuse pattern rather than the typical foci on DNA lesions ? Perhaps also the quantification of Figure 4B should be repeated by separating cells with foci vs cells with diffuse staining of phospho-ATM ? It could also be interesting to inhibit ATM kinase activity and see whether this decreases inter-cellular NMD variability.

3-In Figure 3B and 4B, perhaps indicate the line and Pearson coefficients directly in the Figure panels.

Reviewer #3 (Remarks to the Author):

This an interesting and important paper that utilizes state-of-the-art technologies to demonstrate that not all cells support NMD with equal efficiencies. The authors show that efficiencies correlate with SMG1 levels (less SMG1 leads to less-efficient NMD and less p-UPF1) and p-ATM S1981 levels (less p-ATM S1981 and a more diffuse staining of nuclei correlates with less efficient NMD).

The paper would be greatly enhanced by clearer writing, quantitations of the level of differences in the abundance of SMG1 and p-ATM S1981 that can be associated with efficient or inefficient NMD, and how the authors rationalize their obtaining a remarkably large percent (20%) of ribosomes reading-through their PTC.

The authors might think about removing the data demonstrating that SMG1 protein levels do not correlate with SMG1 mRNA levels since they do not have a mechanistic explanation for this finding. The proposal that it is autoregulation would apply to the other NMD factors that are likewise encoded by NMD targets, so that proposal requires fleshing out to have any real meaning.

Specific comments to be addressed:

In Figure 1G, what percentage of cells fail to trigger the NMD of the PTC-containing mRNA?

In Figure 3, it is difficult to determine the relative range of SMG1 protein that supports NMD vs doesn't support NMD. It would be helpful if the authors would specify the fold-difference with +/- variability. It would also be helpful for the authors to comment on the considerable variation in the level of FP-WW signal.

Please specify in the text how phosphorylated UPF1 levels are measured – what phosphorylation site(s) are quantitated?

In the text, the authors note that the level of ATM S1981 phosphorylation (and not ATM per se) and its change from nuclear puncta to a homogenous distribution was found in cells that escaped NMD. The authors conclude that stress in response to DNA damage correlates with NMD efficiency, which would be clearer if the authors replaced “NMD efficiency” with “active NMD” since efficiency could be either up or down. The authors write, “However, the level of ATM S1921 phosphorylation did correlate with NMD efficiency and it was translocated to the nucleus in escaped cells.” However, it looks from Figure 4A that p-ATM is always in the nucleus, so the translocation statement is also very confusing.

On a related issue, one would think that diffuse p-ATM S1981 staining throughout nuclei would indicate that there is little dsDNA break repair ongoing since doesn't localization to puncta indicate localization to sites of DNA repair? Some explanation of the functional difference between diffuse nuclear localization and localization in nuclear puncta seems important.

Regarding Figure 5, one would think that cells would vary in their degree of PTC readthrough. Thus, stating in the text that “80% of translating ribosomes successfully terminated at the PTC, and 20% engaged in translational readthrough during NMD escape” is remarkable not only in the degree of readthrough but also in the uniformity of readthrough efficiency, it would seem.

The authors state that NMD efficiency may be transcript-specific. This seems to be an important topic for the Discussion since the authors measured only one PTC. Do the authors have any insight into whether the percentage of readthrough translation is the same for all three nonsense codons? One wonders about the latter since one mechanism of escape could be nonsense suppression, and the efficiency of suppression could conceivably be different for different termination codons at the same position in a transcript (let alone the position of a termination codon within the open reading frame and other features of particular transcripts).

The authors note that the decreased level of SMG1 in cells that escape NMD vs cells that have active NMD is not accompanied by a decreased level of SMG1 mRNA but, in fact, an increased level of SMG1 mRNA. The authors do not have a very good explanation of this, especially since the mRNAs for other NMD factors also derive from NMD targets. How do the authors believe these data add to their manuscript?

The Discussion could be much more informative, focusing on what the authors learned from their experiments rather than spending so much time reviewing the technology. The comment about their measurements occurring in translationally active cells is important and they do make it. However, they could make other important conclusions/comments, driving home how their findings have added to what is known.

Comment about manuscript formatting: In the future, please number pages and lines on pages, double-

spaced, so reviewers can reference a particular spot in the manuscript. The format of this manuscript is very unusual and difficult for reviewers to work with.

Response to Reviewers

REVIEWER COMMENTS

Reviewer #1 (Remarks to the Author):

Sato and Singer use a single-cell approach with a bi-directional reporter to demonstrate an important concept for the RNA field: dramatic cell-to-cell variability of NMD efficiency. With regard to mechanism, they used a fluorescent reporter system coupled with FACS to reveal that cells that undergo NMD escape do this via translational readthrough of the premature termination codon (PTC) or via failure to trigger the mRNA degradation mechanism after successful translation termination at the PTC. While they did not define the precise underlying mechanism responsible, they did show that NMD escape correlates with low levels of the NMD kinase, SMG1, and phosphorylated UPF1. Overall, the manuscript is interesting and the results are convincing. That said, this manuscript is weak with regard to depth, and there are some other concerns:

We thank Reviewer #1 for carefully reading the review and the useful comments, suggestions and corrections. We introduced all the modifications suggested by Reviewer #1.

1. Depth. The authors make various tantalizing discoveries but go no further with them. It is critical that they go more in depth with just ONE of these discoveries. For example, the authors found a correlation between ATM autophosphorylation and NMD escape (Fig. 4A), which led them to speculate that NMD efficiency is linked with DNA damage responses. The authors could examine this DNA damage hypothesis in more detail. Although causal mechanism would be preferred, they could at least perform a correlative study examining other DNA-damage-associated molecules for their association with NMD-escape. A second avenue they could go down is to investigate molecularly what might distinguish NMD-escape vs. NMD-active cells. For example, the authors could do RNA-seq analysis to investigate—at least at the correlative level—what transcripts are associated with these two phenotypes. A third discovery the authors could follow up on is how cells escape NMD despite apparently terminating translation at the PTC (80% of translating ribosomes). The authors may have other ideas on how to enhance depth; the point is to go deeper on one sub-topic to bring this paper to the level of this journal.

As suggested, we examined further investigation towards the reduction of SMG1 under DNA damage response using drugs that induce DNA damage in new Figure 4 and showed that doxorubicin treatment significantly reduced the level of SMG1. Unfortunately, we could not test the NMD efficiency with doxorubicin treatment due to its intrinsic fluorescence. Instead, we showed that a high concentration of etoposide, which resulted in the mild reduction of SMG1, increased NMD escape.

2. The following sentence is vague: "The interplay between SMG1 and ataxia-telangiectasia mutated (ATM) protein..." (in the last paragraph in page 3). Are the authors stating there is an interplay between these molecules? If so, what specifically is the nature of this interplay?

We agree with reviewer #1 that this sentence is not accurate. Now we have modified the sentence as "SMG1 exhibits some functional overlap with the other PIKK family member, ataxia-telangiectasia mutated (ATM) protein. Kinase activity in both SMG1 and ATM are stimulated in response to genotoxic stress and phosphorylate their downstream target p53 on serine¹⁵, to coordinate downstream stress-induced signaling pathways".

3. Another sentence in the same paragraph is also vague: "NMD efficiencies may also correlate with ATM

autophosphorylation at Ser 1981.” Do these 2 things correlate or not? If so, please give a reference? If no previous evidence, state as hypothesis or rephrase.

We understand the reviewer’s concern. Since this was mainly conjecture, we removed the correlation result in old Fig. 4B. Instead, we tested the DNA damage inducer to test how the level of SMG1 was altered by DNA damage response in new Figure 4.

4. Rephrase the sentence - “However, the molecular weight of the EGFP-tagged protein and the intensity of band detected...” (second paragraph on page 4) - by omitting "However" and stating as an experiment. Combine this paragraph with the next one, as they are on the same topic.

We have introduced the correction as “The molecular weight of the EGFP-tagged protein in western blotting analysis was used to determine whether the PTC has been readthrough or successfully recognized resulting in the production of full-length or truncated EGFP-tagged proteins”.

5. In the first paragraph of the Discussion, the following is claimed: “...thus it provides the precise detection of NMD efficiency in every cell.” This is an overstatement given the considerable evidence that NMD has different branches coupled with the fact that there is considerable variability as whether a given NMD target mRNA is degraded by NMD in a given cellular context. Thus, it is unlikely that the beta-globin system used by authors will detect “all NMD.” This caveat should be explicitly stated.

The sentence has been removed.

6. The Discussion is highly redundant with the Results. Some reiteration, such as in the first paragraph, is ok, but after that, the focus should be on implications and speculation, not reiteration of the results.

We agree with the reviewer and have removed redundancies.

7. Fig. 1E. Should be changed to 1D so that it is in chronological order in the text.

Now Figure 1E have been changed to 1D as suggested.

8. Fig. 3A. The difference in the IF_SMG1 signal between NMD+ and NMD- cells is not obvious. While the data analysis in panel B demonstrates a convincing modest correlation, more cells should be shown (or some other approach should be used) to visually demonstrate the decrease in SMG1 protein expression in NMD escape cells in panel A.

We have modified figure 3B&C by showing the dataset of NMD emphasized with different colors as well as adding the figure C&D showing the difference in the levels of SMG1 or phosphor-UPF1 between two populations.

9. Consistency in the figures. It is strongly suggested to show significance r and P values in the “correlation” figures themselves (Fig. 3B/C, 4B), as done in Supplemental Figure S5, not in the legend. To reduce space put the r and P values in the upper portion of the figure rather than to the right..

R and P values have been added to the figures as suggested.

Reviewer #2 (Remarks to the Author):

Report on manuscript "Cellular variability of nonsense-mediated mRNA decay", by Hanae Sato and Robert H. Singer.

This manuscript describes the development of a bicolor NMD reporter, based on a bidirectional promoter transcribing both a control and a test NMD RNA. By transiently transfecting human cell lines and measuring mRNA levels in single cells by smFISH, the authors show that NMD efficiency varies quite significantly from cell to cell, with some cells showing little NMD activity. The authors confirm this using GFP/mCherry protein reporters, in the same bidirectional vector, and show that low NMD cells correlated with high SMG1 or phospho-UPF1 protein levels, and also high phospho-ATM. Finally, the authors show that low NMD is due to both PTC readthrough and inefficient mRNA degradation, and that low NMD cells, selected by FACS with the two-color reporter, display widespread NMD defects in endogenous NMD target.

The manuscript is clear and straightforward, and the data are clear cut. The observations are interesting and report on an important but previously overlooked property of NMD. This study will certainly form the basis of many more of the kind to come. I have only few comments and I support publication of the paper provided they are addressed.

We thank Reviewer 2 for the positive comments on our manuscript.

1-All the experiments are done with transient transfections and this requires quite harsh cell treatments possibly generating cell-to-cell heterogeneity (not all cells are transfected). I thus think that it is important that the authors repeat a couple of their key findings using a stable cell line. I am thinking to Figure 1 of course, but also Figure 5.

We have established the Flp-In fluorescence-NMD reporter (For Figure 1 and 5) and the figure now has been added in Figure S6.

2-In the experiment with phospho-ATM (Figure 4), I am not sure I get the point with the diffuse staining. Could the authors elaborate on what causes this diffuse pattern rather than the typical foci on DNA lesions ? Perhaps also the quantification of Figure 4B should be repeated by separating cells with foci vs cells with diffuse staining of phospho-ATM ? It could also be interesting to inhibit ATM kinase activity and see whether this decreases inter-cellular NMD variability.

We agree with the reviewer's concern. We removed the correlation result in old Fig. 4B. Instead, we tested the DNA damage inducer to test how the level of SMG1 was altered by DNA damage response in new Figure 4.

3-In Figure 3B and 4B, perhaps indicate the line and Pearson coefficients directly in the Figure panels.

They have been added to the figures as suggested.

Reviewer #3 (Remarks to the Author):

This an interesting and important paper that utilizes state-of-the-art technologies to demonstrate that not all cells support NMD with equal efficiencies. The authors show that efficiencies correlate with SMG1 levels (less SMG1 leads to less-efficient NMD and less p-UPF1) and p-ATM S1981 levels (less p-ATM S1981 and a more diffuse staining of nuclei correlates with less efficient NMD).

The paper would be greatly enhanced by clearer writing, quantitations of the level of differences in the abundance of SMG1 and p-ATM S1981 that can be associated with efficient or inefficient NMD, and how the authors rationalize their

obtaining a remarkably large percent (20%) of ribosomes reading-through their PTC.

The authors might think about removing the data demonstrating that SMG1 protein levels do not correlate with SMG1 mRNA levels since they do not have a mechanistic explanation for this finding. The proposal that it is autoregulation would apply to the other NMD factors that are likewise encoded by NMD targets, so that proposal requires fleshing out to have any real meaning.

We thank Reviewer 3 for the positive comment on our manuscript and for suggestions to enhance the significance of our paper. As suggested, we have removed the data showing the SMG1 protein levels do not correlate with SMG1 mRNA levels to avoid confusion.

Specific comments to be addressed:

1. In Figure 1G, what percentage of cells fail to trigger the NMD of the PTC-containing mRNA?

We now have added the histogram in Figure S3 to show what percentage of cells fail to trigger the NMD.

2. In Figure 3, it is difficult to determine the relative range of SMG1 protein that supports NMD vs doesn't support NMD. It would be helpful if the authors would specify the fold-difference with +/- variability. It would also be helpful for the authors to comment on the considerable variation in the level of FP-WW signal.

We have modified figure 3B&C by showing the dataset of NMD escape with different colors as well as adding the figure C&D showing the difference in the levels of SMG1 or phosphor-UPF1 between two populations.

3. Please specify in the text how phosphorylated UPF1 levels are measured – what phosphorylation site(s) are quantitated?

We performed IF using anti- phosphorylated UPF1 antibody (Ser1127 from Millipore Sigma # 07-1016) and measured fluorescence intensities. This is in the text.

4. In the text, the authors note that the level of ATM S1981 phosphorylation (and not ATM per se) and its change from nuclear puncta to a homogenous distribution was found in cells that escaped NMD. The authors conclude that stress in response to DNA damage correlates with NMD efficiency, which would be clearer if the authors replaced “NMD efficiency” with “active NMD” since efficiency could be either up or down. The authors write, “However, the level of ATM S1921 phosphorylation did correlate with NMD efficiency and it was translocated to the nucleus in escaped cells.” However, it looks from Figure 4A that p-ATM is always in the nucleus, so the translocation statement is also very confusing.

We agree with the reviewer's concern. We removed the correlation result in old Fig. 4B. Instead, we tested the DNA damage inducer to test how the level of SMG1 was altered by DNA damage response in new Figure 4.

5. On a related issue, one would think that diffuse p-ATM S1981 staining throughout nuclei would indicate that there is little dsDNA break repair ongoing since doesn't localization to puncta indicate localization to sites of DNA repair? Some explanation of the functional difference between diffuse nuclear localization and localization in nuclear puncta seems important.

We agree with the reviewer's concern. Since the mechanism in translocation of sub-localization of ATM S1981 phosphorylation under DNA damage is not well understood, we removed the correlation assay in old Figure 4 and moved the IF data into supplemental Figure S8.

6. Regarding Figure 5, one would think that cells would vary in their degree of PTC readthrough. Thus, stating in the text that "80% of translating ribosomes successfully terminated at the PTC, and 20% engaged in translational readthrough during NMD escape" is remarkable not only in the degree of readthrough but also in the uniformity of readthrough efficiency, it would seem.

We thank Reviewer 3 for the positive comment regarding Figure 5. We have added a sentence describing this importance in discussion.

7. The authors state that NMD efficiency may be transcript-specific. This seems to be an important topic for the Discussion since the authors measured only one PTC. Do the authors have any insight into whether the percentage of readthrough translation is the same for all three nonsense codons? One wonders about the latter since one mechanism of escape could be nonsense suppression, and the efficiency of suppression could conceivably be different for different termination codons at the same position in a transcript (let alone the position of a termination codon within the open reading frame and other features of particular transcripts).

Our NMD reporter contains the beta globin gene with PTC (UGA codon) and we did not test other termination codons to test the readthrough efficiency which causes NMD escape. It has been shown that the efficiency of translational readthrough is dependent on the termination codons (UAA, UAG, or UGA) and/ or sequence context surrounding of termination codons. Now we include this topic as a potential mechanism of NMD escape in discussion.

8. The authors note that the decreased level of SMG1 in cells that escape NMD vs cells that have active NMD is not accompanied by a decreased level of SMG1 mRNA but, in fact, an increased level of SMG1 mRNA. The authors do not have a very good explanation of this, especially since the mRNAs for other NMD factors also derive from NMD targets. How do the authors believe these data add to their manuscript?

To avoid confusion, we have removed the data as suggested.

9. The Discussion could be much more informative, focusing on what the authors learned from their experiments rather than spending so much time reviewing the technology. The comment about their measurements occurring in translationally active cells is important and they do make it. However, they could make other important conclusions/comments, driving home how their findings have added to what is known.

We thank Reviewer 3 for the comments. We have added sentences to describe further the implications and speculation in discussion.

10. Comment about manuscript formatting: In the future, please number pages and lines on pages, double-spaced, so reviewers can reference a particular spot in the manuscript. The format of this manuscript is very unusual and difficult for reviewers to work with.

We apologize for inconvenience. We have now followed the general manuscript formatting.

REVIEWER COMMENTS

Reviewer #1 (Remarks to the Author):

The authors have responded well to many of the concerns. New data has been added, including data showing that single-cell variation in NMD also occurs in stably transfected cells. This is very important, as it rules that cellular stress caused by transient transfection is responsible for this phenomenon. While there is considerable enthusiasm for this ground-breaking MS, unfortunately, some concerns were either under- or un-addressed:

(1) Rev 1, point 1. It was requested to provide more depth, preferably at the mechanistic level.

Unfortunately, the experiment provided by the authors is underwhelming. The DNA damage agent, etoposide, had no sig effect at low dose, and the higher dose (50 uM) caused an extremely modest reduction in SMG1 level (even though statistically significant). While this high dose did appear to also modestly affect NMD escape, it is not clear what this means. Optimally, it would be best if depth was provided by some other experiment.

(2) Figure placement. Considering that the above new “functional data” is underwhelming, it is not clear why it has replaced previous Fig. 4 as a main figure. It seems better as a supplementary figure.

(3) Rev 1, point 8. The following about Fig. 3A still has still not been addressed: “The difference in the IF_SMG1 signal between NMD+ and NMD- cells is not obvious. While the data analysis in panel B demonstrates a convincing modest correlation, more cells should be shown (or some other approach should be used) to visually demonstrate the decrease in SMG1 protein expression in NMD escape cells in panel A.”

(4) Images. Related to the above, representative images should be provided for Fig. 3d&e and Fig. S7.

(5) Rev 3, introductory comments. It did not appear that the authors addressed the following points: “The paper would be greatly enhanced by clearer writing, quantitations of the level of differences in the abundance of SMG1 and p-ATM S1981 that can be associated with efficient or inefficient NMD, and how the authors rationalize their obtaining a remarkably large percent (20%) of ribosomes reading-through their PTC.”

(6) Rev 3, point 7. The authors nicely responded to the following question: “The authors state that NMD efficiency may be transcript-specific. This seems to be an important topic for the Discussion since the authors measured only one PTC. Do the authors have any insight into whether the percentage of readthrough translation is the same for all three nonsense codons? One wonders about the latter since one mechanism of escape could be nonsense suppression, and the efficiency of suppression could conceivably be different for different termination codons at the same position in a transcript (let alone the position of a termination codon within the open reading frame and other features of particular transcripts).” Authors’ answer: “Our NMD reporter contains the beta globin gene with PTC (UGA codon) and we did not test other termination codons to test the readthrough efficiency which causes NMD escape. It has been shown that the efficiency of translational readthrough is dependent on the termination codons (UAA, UAG, or UGA) and/ or sequence context surrounding of termination codons. Now we include this topic as a potential mechanism of NMD escape in discussion.” While I’m happy with this addition to the Discussion, I think the authors missed an important point: the PTC readthrough they

observed is (I believe) far higher than that previously reported for normal mRNAs. Thus, the authors should first state the PTC readthrough frequency range from past studies, then should speculate why the frequency they observed as higher. Could it be, for example, that past measurements of PTC readthrough may have been inaccurate, as they examined cell populations, not single cells?

(7) Rev 3, point 8. The authors removed the qPCR data showing altered NMD factor mRNA levels in NMD escape cells, based on the following Rev 3 concern: “The authors note that the decreased level of SMG1 in cells that escape NMD vs cells that have active NMD is not accompanied by a decreased level of SMG1 mRNA but, in fact, an increased level of SMG1 mRNA. The authors do not have a very good explanation of this, especially since the mRNAs for other NMD factors also derive from NMD targets. How do the authors believe these data add to their manuscript?” I think the authors’ explanation for this result in their original submission – that it may result from the well-established NMD feedback regulatory loop is perfectly reasonable. Indeed, not only did SMG1 mRNA go up, but so did several other NMD target mRNAs encoding NMD factors: UPF1, SMG5, and SMG7. It is not surprising that some NMD factor mRNAs encoding NMD factors failed to go up, since, as mentioned in the original submission, this feedback regulation has been shown to be tissue specific. Thus, it is strongly suggested that this data be put back in the MS. The upregulation of NMD factors provides important evidence for disrupted NMD efficiency in “NMD escape” cells. In addition, the opposite effect on SMG1 mRNA vs. protein is useful data for the field (indeed, other papers have reported such opposite effects on NMD factor mRNAs vs. proteins).

Reviewer #2 (Remarks to the Author):

In this revised version, the authors have addressed the criticisms of the reviewers and provided new data when necessary. In particular they show that DNA damage agent affect SMG1 levels and increase NMD efficiency variability. The manuscript has improved and I support publication.

In the new Figure S6 with the stable Flp-in cell lines, I suggest that the authors add the Pearson correlation coefficients as a measure of NMD dispersion, as in the main Figures.

Reviewer #3 (Remarks to the Author):

The authors have largely addressed the issues raised. However, a couple of sticking points remain, one generated by the new doxorubicin data.

1. Page 2. The meaning of “a subpopulation of NMD substrates escapes from NMD (~30%)” is unclear. It might be better to state that the efficiency of NMD for many NMD target is ~30%. It also seems important to clarify that some NMD targets undergo very efficient NMD, e.g. the Ig and TCR mRNAs. Since this high efficiency of NMD has been shown for a number of cell types, one wonders how this can be if what the authors are finding is true.

2. Results from the new doxorubicin experiments are confusing in the context of existing literature.

Doxorubicin has been shown to inhibit NMD by the caspase-mediated cleavage of a fraction of cellular UPF1 to generate a functionally dominant-negative cleavage product that, by itself, can inhibit NMD (Popp and Maquat, 2015). Expression of this cleavage product upregulates the level of NMD targets to promote apoptosis since a number of NMD targets encode pro-apoptotic proteins.

How and why do the authors find that doxorubicin decreases the level of SMG1 protein (maybe the level of SMG1 mRNA can be determined in RNA-seq data provided in the Popp and Maquat, 2015 paper), and is this really the cause of NMD escape? Since doxorubicin has been shown to inhibit NMD by production of a dominant-negative product of UPF1, this product could explain the NMD escape the authors observe. The authors note that they cannot measure NMD efficiency due to the high fluorescent background of doxorubicin, thereby obscuring determining cause-and-effect, which seems important to elucidate. Do the authors have some way around this conundrum?

REVIEWER COMMENTS

Reviewer #1 (Remarks to the Author):

The authors have responded well to many of the concerns. New data has been added, including data showing that single-cell variation in NMD also occurs in stably transfected cells. This is very important, as it rules that cellular stress caused by transient transfection is responsible for this phenomenon. While there is considerable enthusiasm for this ground-breaking MS, unfortunately, some concerns were either under- or un-addressed:

(1) Rev 1, point 1. It was requested to provide more depth, preferably at the mechanistic level. Unfortunately, the experiment provided by the authors is underwhelming. The DNA damage agent, etoposide, had no sig effect at low dose, and the higher dose (50 uM) caused an extremely modest reduction in SMG1 level (even though statistically significant). While this high dose did appear to also modestly affect NMD escape, it is not clear what this means. Optimally, it would be best if depth was provided by some other experiment.

We agree that it may not be convincing to show the DNA damage-induced NMD escape since etoposide treatment didn't trigger a strong effect on the escape and we couldn't detect escape with the single-cell approach under doxorubicin treatment which induced reduction of SMG1. Now we have provided additional data using qPCR to show doxorubicin treatment induced the NMD escape.

(2) Figure placement. Considering that the above new "functional data" is underwhelming, it is not clear why it has replaced previous Fig. 4 as a main figure. It seems better as a supplementary figure.

We believe that showing the induction of DNA damage alters the level of SMG1 and induces NMD inhibition is more informative than showing the correlation of the DNA damage marker. We agree that we did not show convincingly the inhibition of NMD under doxorubicin treatment even though we could successfully show the reduction of SMG1 with DNA damage. We now show that NMD escape was increased under doxorubicin treatment using qPCR. We also moved the figures showing etoposide treatment to the supplemental figure S9, and modified the flow cytometry figures with larger dots in figure S9 for clear visualization.

(3) Rev 1, point 8. The following about Fig. 3A still has still not been addressed: "The difference in the IF_SMG1 signal between NMD+ and NMD- cells is not obvious. While the data analysis in panel B demonstrates a convincing modest correlation, more cells should be shown (or some other approach should be used) to visually demonstrate the decrease in SMG1 protein expression in NMD escape cells in panel A."

We now have shown the NMD+ and NMD- cells in the same image to show the clear difference of SMG1 expression.

(4) Images. Related to the above, representative images should be provided for Fig. 3d&e and Fig. S7.

We have provided the images in Fig3d&e and Fig.S7.

(5) Rev 3, introductory comments. It did not appear that the authors addressed the following points: “The paper would be greatly enhanced by clearer writing, quantitations of the level of differences in the abundance of SMG1 and p-ATM S1981 that can be associated with efficient or inefficient NMD, and how the authors rationalize their obtaining a remarkably large percent (20%) of ribosomes reading-through their PTC.”

For the correlation of SMG1 and p-ATM S1981 with efficient or inefficient NMD, we have addressed the quantification of the level of differences in the level of SMG1 in the cell population. For the correlation with p-ATM S1981, we decided to move it to the supplemental just to show the localization differences p-ATM S1981 since this is more informative than correlation of p-ATM S1981 intensities. The answer for the readthrough is described in the next section.

(6) Rev 3, point 7. The authors nicely responded to the following question: “The authors state that NMD efficiency may be transcript-specific. This seems to be an important topic for the Discussion since the authors measured only one PTC. Do the authors have any insight into whether the percentage of readthrough translation is the same for all three nonsense codons? One wonders about the latter since one mechanism of escape could be nonsense suppression, and the efficiency of suppression could conceivably be different for different termination codons at the same position in a transcript (let alone the position of a termination codon within the open reading frame and other features of particular transcripts).” Authors’ answer: “Our NMD reporter contains the beta globin gene with PTC (UGA codon) and we did not test other termination codons to test the readthrough efficiency which causes NMD escape. It has been shown that the efficiency of translational readthrough is dependent on the termination codons (UAA, UAG, or UGA) and/ or sequence context surrounding of termination codons. Now we include this topic as a potential mechanism of NMD escape in discussion.” While I’m happy with this addition to the Discussion, I think the authors missed an important point: the PTC readthrough they observed is (I believe) far higher than that previously reported for normal mRNAs. Thus, the authors should first state the PTC readthrough frequency range from past studies, then should speculate why the frequency they observed as higher. Could it be, for example, that past measurements of PTC readthrough may have been inaccurate, as they examined cell populations, not single cells?

We agree with the reviewer’s comment. We now have added more discussion to explain that we detected a higher level of translational readthrough compared with previous research. We believe that the higher level of translational readthrough detected in our approach is due to sensitive detection in the selective NMD-escape cell population. Notably, we detected this higher readthrough only in the NMD-escape cell population but not in the efficient NMD cell population. This represented the advantage of our single-cell approach,

which provides a sensitive tool to investigate a minor cell population that could not be detected previously.

(7) Rev 3, point 8. The authors removed the qPCR data showing altered NMD factor mRNA levels in NMD escape cells, based on the following Rev 3 concern: “The authors note that the decreased level of SMG1 in cells that escape NMD vs cells that have active NMD is not accompanied by a decreased level of SMG1 mRNA but, in fact, an increased level of SMG1 mRNA. The authors do not have a very good explanation of this, especially since the mRNAs for other NMD factors also derive from NMD targets. How do the authors believe these data add to their manuscript?” I think the authors’ explanation for this result in their original submission – that it may result from the well-established NMD feedback regulatory loop is perfectly reasonable. Indeed, not only did SMG1 mRNA go up, but so did several other NMD target mRNAs encoding NMD factors: UPF1, SMG5, and SMG7. It is not surprising that some NMD factor mRNAs encoding NMD factors failed to go up, since, as mentioned in the original submission, this feedback regulation has been shown to be tissue specific. Thus, it is strongly suggested that this data be put back in the MS. The upregulation of NMD factors provides importance evidence for disrupted NMD efficiency in “NMD escape” cells. In addition, the opposite effect on SMG1 mRNA vs. protein is useful data for the field (indeed, other papers have reported such opposite effects on NMD factor mRNAs vs. proteins).

We put this data back into the manuscript along with a clearer explanation.

Reviewer #2 (Remarks to the Author):

In this revised version, the authors have addressed the criticisms of the reviewers and provided new data when necessary. In particular they show that DNA damage agent affect SMG1 levels and increase NMD efficiency variability. The manuscript has improved and I support publication.

In the new Figure S6 with the stable Flp-in cell lines, I suggest that the authors add the pearson correlation coefficients as a measure of NMD dispersion, as in the main Figures.

We added the Slope, R-square, and Pearson correlation coefficients into the figure legend.

Reviewer #3 (Remarks to the Author):

The authors have largely addressed the issues raised. However, a couple of sticking points remain, one generated by the new doxorubicin data.

1. Page 2. The meaning of “a subpopulation of NMD substrates escapes from NMD (~30%)” is unclear. It might be better to state that the efficiency of NMD for many NMD target is ~30%. It also seems important to clarify that some NMD targets undergo very efficient NMD, e.g. the Ig and TCR mRNAs. Since this high efficiency of NMD has been shown for a number of cell types, one wonders how this can be if what the authors are finding is true.

We have added more references and descriptions showing the differential NMD efficiency in several NMD targets in the sentence “NMD efficiency varies across transcripts (e.g. ~20% in β -globin¹⁰, or Triose phosphate isomerase (TPI) ¹¹, ~4% in T-cell receptor β (TCR- β) ⁹ or Immunoglobulin (Ig) ^{12,13})”, and remove the “(~30%)” to provide the clear statement about differential NMD efficiencies.

2. Results from the new doxorubicin experiments are confusing in the context of existing literature. Doxorubicin has been shown to inhibit NMD by the caspase-mediated cleavage of a fraction of cellular UPF1 to generate a functionally dominant-negative cleavage product that, by itself, can inhibit NMD (Popp and Maquat, 2015). Expression of this cleavage product upregulates the level of NMD targets to promote apoptosis since a number of NMD targets encode pro-apoptotic proteins.

How and why do the authors find that doxorubicin decreases the level of SMG1 protein (maybe the level of SMG1 mRNA can be determined in RNA-seq data provided in the Popp and Maquat, 2015 paper), and is this really the cause of NMD escape? Since doxorubicin has been shown to inhibit NMD by production of a dominant-negative product of UPF1, this product could explain the NMD escape the authors observe. The authors note that they cannot measure NMD efficiency due to the high fluorescent background of doxorubicin, thereby obscuring determining cause-and-effect, which seems important to elucidate. Do the authors have some way around this conundrum?

We initially identified the reduction of SMG1 from the correlation test using the single-cell assay which we developed. Since SMG1 is a family of phosphatidylinositol-3-kinase-like kinases (PIKKs) known to play a major role in the DNA damage response, we tested the novel DNA damage inducers doxorubicin and etoposide, and found both reduced the levels of SMG1. However, we agree that it was not convincing without showing the inhibition of NMD under doxorubicin treatment. We now have shown that NMD escape is increased (NMD inhibition) under doxorubicin treatment using qPCR to confirm the increase of NMD escape under doxorubicin treatment.

As reviewer 3 mentioned, doxorubicin-induced NMD inhibition is consistent with the previous research (Popp and Maquat 2015) and it is not clear that NMD inhibition under doxorubicin treatment was caused by DNA damage or apoptosis since they are tightly linked. We now have included this point in the discussion.

REVIEWERS' COMMENTS

Reviewer #1 (Remarks to the Author):

I'm happy to say that the authors have done an excellent job of revising their MS. I'm satisfied.